# Two-Dimensional Mammography Imaging Techniques for Screening Women with Silicone Breast Implants: A Pilot Phantom Study

**DOI:** 10.3390/bioengineering11090884

**Published:** 2024-08-31

**Authors:** Isabelle Fitton, Virginia Tsapaki, Jonathan Zerbib, Antoine Decoux, Amit Kumar, Aude Stembert, Françoise Malchair, Claire Van Ngoc Ty, Laure Fournier

**Affiliations:** 1Department of Radiology, AP-HP, Hôpital Européen Georges Pompidou, 75015 Paris, France; jonathan.zerbib@aphp.fr (J.Z.); claire.vanngocty@aphp.fr (C.V.N.T.); 2Division of Human Health, Department of Nuclear Sciences and Applications, International Atomic Energy Agency, 1220 Vienna, Austria; v.tsapaki@iaea.org; 3Paris Cardiovascular Research Center, Institut National de la Santé et de la Recherche Médicale Unité 970, 75015 Paris, France; 4OKOMERA, iPEPS, The Healthtech Hub, 75013 Paris, CEDEX 13, France; aamitce@gmail.com; 5ZEPHYRA, 4000 Liège, Belgium; aude.stembert@zephyra.be (A.S.); francoise.malchair@zephyra.be (F.M.); 6Department of Radiology, PARCC UMRS 970, INSERM, Hôpital Européen Georges Pompidou, Université Paris Cité, AP-HP, 75015 Paris, France; laure.fournier@aphp.fr

**Keywords:** breast, breast implant, mammography, quality control, ionizing radiation, education

## Abstract

This study aimed to evaluate the impact of three two-dimensional (2D) mammographic acquisition techniques on image quality and radiation dose in the presence of silicone breast implants (BIs). Then, we propose and validate a new International Atomic Energy Agency (IAEA) phantom to reproduce these techniques. Images were acquired on a single Hologic Selenia Dimensions^®^ unit. The mammography of the left breast of a single clinical case was included. Three methods of image acquisition were identified. They were based on misused, recommended, and reference settings. In the clinical case, image criteria scoring and the signal-to-noise ratio on breast tissue (SNR_BT_) were determined for two 2D projections and compared between the three techniques. The phantom study first compared the reference and misused settings by varying the AEC sensor position and, second, the recommended settings with a reduced current-time product (mAs) setting that was 13% lower. The signal-difference-to-noise ratio (*SDNR*) and detectability indexes at 0.1 mm (d’ 0.1 mm) and 0.25 mm (d’ 0.25 mm) were automatically quantified using ATIA software. Average glandular dose (AGD) values were collected for each acquisition. A statistical analysis was performed using Kruskal–Wallis and corrected Dunn tests (*p* < 0.05). The SNR_BT_ was 2.6 times lower and the AGD was −18% lower with the reference settings compared to the recommended settings. The SNR_BT_ values increased by +98% with the misused compared to the recommended settings. The AGD increased by +79% with the misused settings versus the recommended settings. The median values of the reference settings were 5.8 (IQR 5.7–5.9), 1.2 (IQR 0.0), 7.0 (IQR 6.8–7.2) and 1.2 (IQR 0.0) mGy and were significantly lower than those of the misused settings (*p* < 0.03): 7.9 (IQR 6.1–9.7), 1.6 (IQR 1.3–1.9), 9.2 (IQR 7.5–10.9) and 2.2 (IQR 1.4–3.0) mGy for the *SDNR*, d’ 0.1 mm, d’ 0.25 mm and the AGD, respectively. A comparison of the recommended and reduced settings showed a reduction of −6.1 ± 0.6% (*p* = 0.83), −7.7 ± 0.0% (*p* = 0.18), −6.4 ± 0.6% (*p* = 0.19) and −13.3 ± 1.1% (*p* = 0.53) for the *SDNR*, d’ 0.1 mm, d’ 0.25 mm and the AGD, respectively. This study showed that the IAEA phantom could be used to reproduce the three techniques for acquiring 2D mammography images in the presence of breast implants for raising awareness and for educational purposes. It could also be used to evaluate and optimize the manufacturer’s recommended settings.

## 1. Introduction

Breast cancer is the most frequently diagnosed cancer and the leading cause of cancer death worldwide, with epidemiological studies reporting a steady increase in its incidence [1,2]. A mammogram is the first-line examination used for the detection of breast abnormalities, including benign and/or malignant lesions. Its main role is to enable the detection of breast cancer earlier than clinical examinations [3,4]. Mammography imaging has evolved in recent years, particularly with the introduction of digital technology [5]. Modern digital X-ray mammography systems also have post-processing tools to image breasts with implants more efficiently than in the past. In recent years, breast implants have become a very important issue in most countries as cosmetic and reconstructive procedures increase [6,7]. However, in mammography, it is still a challenging process. In fact, image quality is negatively affected by the radiopaque properties of breast implants, the limitations of breast compression, and the reduction in the areas of imaged breast tissue [8,9]. There are methods to overcome these issues, such as using views with the back placement of the implant against the chest wall [10]. However, there are no guidelines available about mammography acquisition techniques [11] or widely agreed image quality criteria [12,13] for women with breast implants. One difficulty is the variety of breast implant compositions. Breast implants can be filled with a wide range of materials: normal saline, triglyceride solution, silicone gel, polyvinylpyrrolidone solution in saline or glycerin saline with different physical properties. Individual implants therefore exhibit very different energy absorptions [14]. Young et al. [14] showed that triglyceride solution is forty-five times more radiolucent than silicone gel. And even with silicone gel, its properties vary depending on the implant and manufacturer [15]. One of the most common materials used in breast implants is silicone due to its inherent properties including its ease of fabrication, oxygen permeability, flexibility and low cost [16].

The accurate performance of mammography equipment is critical. It has been clearly demonstrated that the ability to detect breast lesions and cancer earlier is compromised when mammography image quality is inadequate [17]. Since the technique also uses ionizing radiation for image production, it is important that it is optimal [18,19]. This is particularly important for mammography, which is used in asymptomatic women as part of screening programs [20]. Therefore, the benefit/risk balance must be ensured. There are a number of international guidelines that provide guidance to users on quality control (QC), quality assurance and dosimetry in mammography [21,22,23,24,25,26,27,28,29]. Acknowledging the need for high-quality and accurate mammography equipment, the International Atomic Energy Agency (IAEA) recently released a remote and automated daily and weekly QC methodology [21,30]. This includes instructions for constructing a simple phantom and free software for image quality and dose assessments [30]. The phantom, which is simple in design and inexpensive to manufacture, enables the advanced evaluation of mammography image quality and is accessible to low-income countries whose breast cancer incidence rates are steadily increasing [1]. Some of the measurements facilitate the tracking of system performance over time, whereas others such as the detectability index (d’) can be used for quality benchmarking between systems [31,32]. 

In order to investigate whether the IAEA methodology could be implemented in radiology centers worldwide across diverse radiological settings, the IAEA recently launched a coordinated research project entitled “Advanced Tools for Quality and Dosimetry of Digital Imaging in Radiology” [33]. Within this context, an individual research project on breast implant (BI) imaging was initiated with the following main objectives: first, to illustrate the different techniques of 2D mammography image acquisition in presence of BIs in a clinical case; second, to evaluate how silicone BI radiopacity may affect the image quality and radiation dose depending on the technique used; and third, to explore whether and how IAEA tools and procedures could assist in the selection of the most appropriate X-ray beam settings by simulating and comparing the different acquisition techniques. 

## 2. Materials and Methods

### 2.1. Clinical Case Illustrating the Different Acquisition Techniques

This retrospective study was conducted within the radiology department of the European Georges Pompidou hospital (Paris, France) accredited for breast cancer screening. Only one particular clinical case was retrospectively selected to illustrate the clinical scenarios of mammography with BIs.

#### 2.1.1. Clinical Context and Characteristics of Breast Implants

Two 2D mammograms of a 47-year-old woman were retrospectively studied. The patient underwent her annual follow-up after breast reconstruction in 2021 and 2022. This patient has a history of grade-3 right triple-negative breast cancer, which is hormone receptor- and HER2-negative. She was treated by a total and partial mastectomy of her right and left breasts, respectively, radiation therapy and hormone therapy. 

The patient had bilateral breast reconstruction using anatomic BIs. They were characterized by an outer membrane of macro textured silicone. They contained a cohesive silicone gel (Groupe Sebbin SAS, Paris, France, Sebbin LSA SM370). These BIs had the following characteristics: base, height, highest projection point, arc length and volume of 125 mm, 115 mm, 52 mm, 71 mm and 370 mL, respectively. The patient provided written consent for use of her imaging data.

#### 2.1.2. Follow-Up by Mammography Imaging

After her breast reconstruction, the patient was monitored by the medical team. In particular, she underwent two 2D mammograms a year apart between 2021 and 2022. Mammography image acquisition for patients with breast implants should follow a specific protocol. However, the distinguishing feature of the mammograms acquired in this clinical case study is that the images were not acquired using the same protocol. In 2021, images were acquired using manual kV_p_ and mAs parameters selected by the technologist. In 2022, another technologist acquired the images in fully automatic mode. 

### 2.2. Digital Mammography System

The conventional 2D full-field digital mammography system used in this work was a Hologic Selenia Dimensions^®^ unit (Hologic Inc., Bedford, MA, USA; Software version: v1.11.0.8). 

This unit has an amorphous selenium detector with an area of 24 × 29 cm^2^ and 70 μm pixel size. The direct current (DC) offset added to the detector signal is equal to 50. The signal-to-noise ratio (SNR) values obtained from the statistics dialog box on the acquisition console take this DC offset into account.

The mammography device complied with national regulations [34] and its performance was within international standards [5,19,22]. Breast average glandular dose (AGD) was calculated according to the EUropean REFerence (EUREF) protocol [35]. Image parameters for acquisition (kVp, mAs, breast thickness, force and anode/filter) and displayed AGDs were collected for each acquisition (Table 1). 

Special implant processing is available on the acquisition console. It must be activated prior to image acquisition.

### 2.3. Characterization of the Automatic Exposure Control

#### 2.3.1. Description of the Automatic Exposure Control

The automatic exposure control (AEC) sensor on this mammography unit is movable. It consists of seven positions marked 1 through 7 on the compression paddle. The first position is the closest to the chest wall boundary and the seventh is the furthest (Figure 1). In the case of automatic exposure, the second position is selected by default by the mammography unit. The position of the AEC sensor according to the BI can be displayed and recorded on the workstation.

#### 2.3.2. Tests of Automatic Exposure Control Sensitivity

In order to evaluate the AEC sensitivity, a flat-field image was produced for each of the seven positions of the AEC sensor to evaluate spatial variations due to imaging system properties. 

A 4.0 cm thick uniform slab of poly(methyl methacrylate) (PMMA) covering the entire detector was imaged. 

Acquisitions were repeated five times, using a tungsten (W)/rhodium (Rh) target/filter combination and the compression paddle. The variation in kVp, mAs and compressed thickness, as well as displayed AGDs were tracked for each acquisition. 

Five 256 × 256 regions of interest (ROIs) were used to determine the signal-to-noise ratio (SNR). The ROIs were positioned at the center (C) and at the four corners of the images (CW1, CW2, N1 and N2) (Figure 1a). SNR variation was evaluated from the center to the four corners of the image. Coefficients of variation (COVs) were calculated for each parameter to evaluate its accuracy and reproducibility.

### 2.4. Description of the IAEA Phantom for Mammography

The IAEA methodology was followed for phantom construction, image quality and dose evaluation [21].

The phantom was constructed in-house. It was composed of a 24 × 30 × 4.5 cm^3^ plate of PMMA serving as a uniform attenuator, a 5 × 5 cm^2^ copper (Cu) square with a thickness of 0.1 cm and a 1 × 1 cm^2^ aluminium (Al) square with a thickness of 0.02 cm (Figure 2). A thickness of 4.5 cm of homogeneous PMMA without Cu is equivalent in terms of absorption to a breast thickness of 5.3 cm without BIs [35]. Although PMMA is not an exact tissue substitute, it represents an approximation of average breast absorption that is commonly used for quality control purposes [35]. 

The use of copper in mammography is already known in the form of self-adhesive tape. It can be used to support the radiopaque markers chosen to keep the marker information visible over a wide range of exposures [36]. 

### 2.5. Description of 2D Mammographic Imaging Techniques for Breast Implants and Assimilates

#### 2.5.1. Image Acquisition Techniques in Patients with Breast Implants

A team of radiologists and technologists performed the mammograms for the clinical case, including 2D projections of each breast: craniocaudal (CC) and medio-lateral oblique (MLO). 

The technologist can perform the mammography exam using two modes: 

First, if the BI can be moved, the Eklund maneuver can be applied. After a total mastectomy, when the breast implant fills the entire breast, it is not possible to perform the Eklund maneuver. This maneuver should be used during CC projection [37]. This technique involves placing the implant directly against the chest wall and then pulling the breast tissue in front of it. This method applies whether the implant is positioned behind or in front of the pectoral muscle, as long as the implant stays soft and unencapsulated. This allows more breast tissue to be shown on the image. In this case, images can be acquired using automatic parameters, as in standard mammograms, with the AEC sensor position on the breast tissue. These automatic settings, without positioning the AEC sensor on the implant, were used as a reference in our study. A misused situation was defined as the partial or complete placement of the AEC sensor on the BI in automatic mode (Figure 3).

Second, the BI cannot be moved and may cover a large portion or all of the breast tissue. In this situation, the manufacturer recommends using manual parameters for acquisition without accounting for the differences in BI material. This ensures that the AEC sensor does not focus on the BI.

#### 2.5.2. Replication of Clinical Acquisition Techniques with the IAEA Phantom

##### Image Acquisition in Automatic Mode with Various Positions of the AEC Sensor

In the clinical situation and automatic mode, the AEC sensor can be placed outside the BI or below the BI, either partially or completely (Figure 3). These situations were simulated using the IAEA phantom by manually moving the AEC sensor from the first to the seventh position, as indicated on the chest compression paddle (Figure 1b). Tube voltages, mAs and target/filter combinations were automatically selected by the system according to phantom thickness and AEC sensor position. 

Case with AEC sensor partly positioned below the copper plate:

From the first to the third position, the AEC sensor was partially under the Cu plate. Positions 1 and 2 were more obscured by the Cu plate than position 3 (Figure 1b). This case was defined as a misused situation.

Case with AEC sensor positioned outside the copper plate:

Starting from the fourth position, the AEC sensor was totally outside the Cu plate. The image quality and dose at the AEC sensor positions outside the Cu plate were defined as references for optimal image acquisition. This was similar to the situation when an Eklund maneuver was performed.

##### Image Acquisition According to Several Settings in Manual Mode

Since the mAs value had to be adjusted manually when the Eklund maneuver was not possible, image quality and radiation dose were compared between several values. The first mAs value was set according to the manufacturer’s recommendations, i.e., 120 mAs for a breast thickness between 4 cm and 6 cm. The second mAs value was defined by selecting the manufacturer’s default mAs reduction option that is available on the acquisition console. This option allows the operator to decrease the mAs values according to determined mAs levels. The first level of mAs reduction was used to assess how image quality performance would degrade. These results were compared with the reference data defined in the previous section, corresponding to the AEC sensor positioned outside the Cu plate.

For both modes, each acquisition was repeated ten times over three weeks to check short-term reproducibility by a medical physicist. The compression force was 50 N. All exposure parameters were recorded. 

### 2.6. Image Quality Evaluation of Breast Implants and Assimilates

#### 2.6.1. Image Quality Evaluation in Patients with Breast Implants

The image quality was retrospectively evaluated by one breast radiologist with 5 years of experience blinded to AEC sensor positioning. Image quality criteria was grouped into three items (positioning, artefacts and sharpness) to assess mammography examinations, as described in previous studies [38,39,40,41]. Each criteria was evaluated for each projection as “correct”, “incorrect” or “not applicable”.

Five 256 × 256 ROIs were used to determine the SNR of breast tissue (SNR_BT_). The ROIs were positioned anteriorly and laterally to the implant. The SNR_BT_ was obtained from the statistics dialog of the acquisition console. 

#### 2.6.2. Image Metrics and Task-Based Image Quality Assessment of the IAEA Phantom

Image quality evaluation and task-based image quality assessment were performed using the ATIA software developed by the IAEA from “for processing” images [30].

Signal-Difference-to-Noise Ratio

By positioning two ROIs, one on the Al square and one on the local background, the signal-difference-to-noise ratio (*SDNR*) was determined as follows (Equation (1)):(1)$$SDNR=\frac{S_{Background}-S_{Al}}{\sigma_{Background}}$$
where $S_{x}$ is the mean signal in the ROIs, and $\sigma_{Background}$ is the standard deviation in background ROI.

Detectability Index

A non-pre-whitening model observer with eye filter (NPWE) was used to calculate the detectability index (*d’_NPWE_*) (Equation (2)) [42]:(2)$${d^{\prime }}_{NPWE}=\frac{\sqrt{2\pi }C\int_{0}^{\infty }S^{2}\left(u\right)MTF^{2}\left(u\right)VTF^{2}\left(u\right)udu}{\sqrt{\int_{0}^{\infty }S^{2}\left(u\right)MTF^{2}\left(u\right)VTF^{4}\left(u\right)nNPS\left(u\right)udu}}$$
where *u* is the spatial frequency in a visual transfer function (*VTF*), *C* is the contrast measured using the Al square, *S* is the Fourier transform of a disk, *MTF* is the modulation transfer function of the detector before sampling and *nNPS* is the normalized noise power spectrum for the image of interest.

Two task functions assumed to represent circular signals of 0.25 mm and 0.1 mm diameters were simulated, as a reasonable approximation of the smallest microcalcifications. This is a typical and important task for mammography imaging systems [43]. The assumption was that the circular object would have the same contrast as the Al square. The suitability of aluminum as a material to represent microcalcifications has been demonstrated previously [44]. The Fourier transform of this disc-shaped object with a radius of R was a Bessel function of the first order. Using a nominal image magnification of 1.5 and a viewing distance of 400 mm, the eye filter was modelled according to the *VTF*.

Normalized noise power spectrum (*nNPS*) was estimated from a 512 × 512 region in a homogeneous area of the phantom image. Half-overlapping ROIs of 256 × 256 pixels were then extracted for the calculation of the 2D NPS. Using seven spatial frequency bins on either side of the axes of the axial spectra, the axial and radial NPS curves were sectioned from the 2D spectra. The *nNPS* was calculated using a standard formula [45,46].

The presampled modulation transfer function (*MTF*) was measured from the edges of the Cu plate in the phantom. The directional *MTF* was obtained at highly supersampled pseudo-frequencies. These were produced by the sloping horizontal and vertical edges. The two orthogonal *MTF* curves were then averaged and evaluated at the same frequencies as the *nNPS*. 

### 2.7. Data Analysis

A statistical approach was used for the interpretation of the image metrics obtained from the IAEA phantom with the different acquisition methods: First, the effect of the AEC sensor position on the image quality was compared between the reference setting and the misused setting; second, the image quality performances of the recommended and optimized manual mAs settings were evaluated against the reference automatic mAs setting.

Preliminary analysis, using the Shapiro–Wilk [47] test and Levene [48] test, was performed in order to verify the normal distribution of data. As most of the data did not conform to a strict normal distribution and some groups exhibited uneven variances, a non-parametric Kruskal–Wallis test [49] was applied. 

In addition, post-hoc comparisons were made using Dunn’s test [50]. *p*-values were reported after applying Bonferroni correction. For all tests, the significance level was set to 0.05 [51]. The data analysis was performed using Python (version 3.9.12), along with the scipy (version 1.7.3) and scikit_posthocs libraries (version 0.7.0). All data were presented as median (interquartile range (IQR)), except when only five data were examined, in which case data were reported as mean ± standard deviation (SD).

## 3. Results

### 3.1. Clinical Case Illustrating the Different Acquisition Techniques

To illustrate the different techniques for acquiring 2D mammographic images in the presence of breast implants, the case study data excluded the patient’s right breast. Indeed, the patient had undergone a total mastectomy of this breast. As a result, one of the automatic image acquisition techniques associated with the Eklund maneuver was not feasible. The case study therefore involved only data on the acquisition of the left breast.

The different image acquisition scenarios in the presence of a breast implant were illustrated by analyzing the data from the left breast between 2021 and 2022 for each projection. The silicone implants were seen as dense ovoid masses and did not show a separate envelope in the mammogram (Figure 4). The left medio-lateral oblique (LMLO) and the left cranio caudal (LCC) projections were included in this analysis. 

Table 1 shows the manual and fully automatic AEC acquisition settings for these two projections. Table 2 shows the evaluation of the image quality by the radiologist.

#### 3.1.1. Impact of Breast Implant Radiopacity According to Acquisition Technique Based on the Recommended Settings Compared to the Reference Settings 

The Eklund maneuver was implemented for the LCC projection of the examinations performed in 2021 and 2022 (Figure 4a,b). In manual mode, the acquisition parameters recommended by the manufacturer were applied. In automatic mode, the parameters were automatically selected according to the position of the AEC cell. This position is shown as a square in Figure 4b and located on the breast tissue.

When applying the automatic settings, the reference AGD was −18% lower than that when using the recommended settings. It decreased from 1.7 mGy to 1.4 mGy (Table 1). The reference SNR_BT_ was 2.6 times lower than that when using the recommended settings. 

Table 2 shows there was no clinically relevant difference in the image quality of the 2D mammograms between the recommended and reference settings, even if some criteria were not fulfilled regarding the positioning according to the recommended settings, such as “nipple in profile or transected by skin” and “visibility of implant edge in the image” for the LCC projection. The clinical value of the mammography examination was considered to be maintained despite these unfulfilled criteria.

**Figure 4 bioengineering-11-00884-f004:**
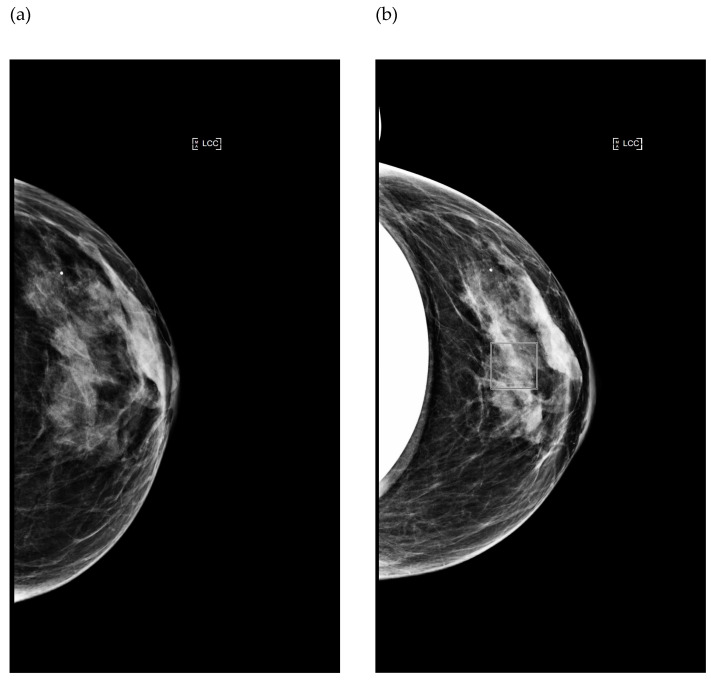
Image comparison between two different mammograms performed in the same patient during follow-up, using the manual (**a**,**c**) and fully automatic (**b**,**d**) modes. (**a**,**b**) Left cranio caudal projection; (**c**,**d**) Left medio-lateral oblique projection. The squares displayed on images indicate the location of the AEC sensor for the automatic mode.

#### 3.1.2. Impact of Breast Implant Radiopacity According to Acquisition Technique Based on the Recommended Settings Compared to the Misused Settings 

Figure 4c,d show the LMLO projection (Figure 4c,d) in the manual and automatic settings. In manual mode, the parameters recommended by the manufacturer were used. In automatic mode, as shown in Figure 4d by the square that represents the position of the AEC, the AEC partially interfered with the breast implant: this was a misuse of the image acquisition technique. The breast tissue was overexposed but analyzable. A surgical marker is apparent on the images.

Regarding the radiation dose, a large increase from 1.4 mGy to 2.5 mGy (+79%) in AGD was observed with the misused settings compared to the recommended settings (Table 1). SNR_BT_ values increased by +97.7% in the misused compared to the recommended settings. We hypothesized that the automatic placement of the AEC sensor partially over the BI was responsible for this increase (Figure 4d), as well as the difference in automatic kVp selection. With regard to this last point, it should be noted that breast compression was less severe in the case of misused settings: 66 mm instead of 51 mm.

Table 2 shows that there was no clinically relevant difference in 2D mammography image quality between the manual and automatic modes, despite the AEC sensor interfering with the BI.

### 3.2. Digital Mammography System: Tests of Automatic Exposure Control Sensitivity

Table 3 shows the impact of the AEC sensor position on the acquisition parameters, AGD and SNR. For all AEC sensor positions and all tracked parameters, the COVs were less than 0.02. The SNR values at the center and four corners of the images were very stable as the AEC sensor position varied, with COVs of less than 0.02. (Table 3). When the mean SNR_C_ was compared to the mean SNR_CW1_, mean SNR_CW2_, mean SNR_N1_ and mean SNR_N2_, the variation was +4.2 ± 0.0%, +6.7 ± 0.0%, −28.3 ± 0.3% and −25.7 ± 0.3%, respectively.

### 3.3. Replication of Clinical Acquisition Techniques with the IAEA Phantom

#### 3.3.1. Comparison of Acquisition Techniques in Automatic Mode: Reference versus Misused Settings with Various Positions of the AEC Sensor

The variation in image quality parameters and AGD for several AEC sensor positions relative to the Cu plate is depicted in Figure 5 for the seven AEC sensor positions. 

As shown in Figure 5a–c, the *SDNR* and detectability indices were lowest when the AEC sensor was located outside the Cu plate with the reference settings. The *SDNR* values significantly decreased when the AEC sensor was out of the Cu plate compared to when the AEC sensor interacted with the Cu plate: −26 ± 6% (*p* ≤ 0.03). The detectability values at 0.1 mm and 0.25 mm were significantly lower when the AEC sensor was outside the Cu plate than when it was partially positioned on the Cu plate: −24 ± 4% (*p* ≤ 0.03) and −23 ± 5% (*p* ≤ 0.03) for d’ (0.1 mm) and d’ (0.25 mm), respectively.

As shown in Figure 5d, the mAs values were statistically comparable between positions 1 vs. 2 (*p* = 1.00), 1 vs. 3 (*p* = 0.64) and 2 vs. 3 (*p* = 1.00) with a median of 173.5 (63.5) (Figure 5d). Also, the mAs values for positions 4 vs. 5, 4 vs. 6, 4 vs. 7, 5 vs. 6, 5 vs. 7 and 6 vs. 7 showed no statistical difference (*p* = 1.00), and the median value was 95.0 (IQR 93.0–97.0). The mAs values for positions 1, 2 and 3 when the AEC sensor was positioned over the Cu plate were significantly higher than the mAs values for positions 4, 5, 6 and 7 when the AEC was positioned outside the Cu plate: +83 ± 32%; (*p* ≤ 0.03). When the AEC sensor was positioned over the copper plate, the mammography unit sought to compensate the photon flux by increasing the mAs values, thereby impacting all the image quality metrics. We found that the kVp values remained stable at 28 kVp (Figure 5f).

As can be seen in Figure 5e, the AGD was at its highest when the AEC sensor was located under the Cu plate and closest to the chest wall boundary at AEC position 1. Although the AGD values appeared different due to positions 1 and 2 being more obscured by the Cu plate than position 3, no statistical difference was observed between positions 1 vs. 2 (*p* = 1.00), 1 vs. 3 (*p* = 0.64) or 2 vs. 3 (*p* = 1.00). The median values were 2.18 (IQR 1.39–2.97) mGy. Conversely, when the AEC sensor was removed from the Cu plate, the radiation doses were 1.18 (IQR 1.15–1.21) mGy and the lowest. They were statistically similar between positions 4, 5, 6 and 7 (*p* = 1.00). For the AEC positions greater than or equal to 4, the AGD values were significantly lower than the AGD values for AEC positions 1, 2 and 3, which were −82 ± 32% (*p* ≤ 0.03) (Table 4). 

#### 3.3.2. Comparison of Several Settings in Manual Mode

Image quality and AGD performances were compared for the manufacturer’s recommended, reduced and reference mAs values (Figure 6). Note that the AGD values obtained from the manufacturer’s recommended mAs for the clinical case and the phantom were qualitatively close to 1.5 (IQR 1.4–1.6) mGy (Table 1 and Figure 6). 

When the mAs values were reduced, the image quality parameter values compared to the recommended mAs values were −6.1 ± 0.6% (*p* = 0.83), −7.7 ± 0.0% (*p* = 0.18) and −6.4 ± 0.6% (*p* = 0.19) for *SDNR*, with detectability at 0.1 mm and 0.25 mm, respectively. The decrease in AGD was −13.3 ± 1.1% (*p* = 0.53). There was no statistical difference found in image quality between the recommended vs. optimized mAs levels (*p* = 0.47); however, the reduced manual exposure for this experiment had a 13% reduction in AGD. 

The comparison between the reduced and reference values showed that all the image metrics and radiation dose data were significantly different (*p* < 0.05). The reference mAs values were significantly lower than the optimized mAs values: −6.8 ± 0.7% (*p* < 0.05). The decrease in all image metrics achieved at reference mAs values compared to reduced mAs values was significantly different: −5.7 ± 0.7% (*p* < 0.05), −5.1 ± 0.7% (*p* < 0.05) and −4.7 ± 0.7% (*p* < 0.05) for *SDNR*, with detectability at 0.1 mm and 0.25 mm, respectively.

When comparing the reference and recommended settings, all image and radiation dose metrics were statistically different (*p* < 0.05). The AGD reference values were lower than the manufacturer’s recommended values for the clinical case and the phantom: −11% and −20 ± 2% (*p* < 0.05), respectively. 

## 4. Discussion

To our knowledge, this is the first mammography study comparing several acquisition techniques in terms of image quality and radiation dose in the presence of BIs or assimilated material on a phantom. In the present study, the IAEA phantom and procedures in mammography were evaluated to determine if and how they could assist in reproducing a misused situation in the presence of a BI and possibly improve the manufacturer’s recommended settings [31]. To this end, an analysis of radiation exposure and image quality in a clinical case was carried out in the presence of the most commonly used silicon-based BI. The use of image quality metrics from the phantom study was not intended to reflect the image quality of a patient’s individual implanted breast. Rather, it was a means of reproducing and comparing the impact of different possible acquisition conditions on image quality and dose in the presence of radiopaque material on the phantom. The clinical case example allowed us to illustrate these different acquisition conditions in manual and automatic modes. All processing of the phantom images was performed automatically using the software developed by the IAEA. This means there will be a better reproducibility of the results compared to visual analysis [9], which may be subject to inter- and intraobserver variabilities [8].

The clinical case analysis showed that without adapted manual settings, the radiation dose can be high without clinical gain in image quality for the radiologist despite a noise reduction. Therefore, incorrect AEC sensor positioning in which it overlaps the BI does not degrade the quality of the exam. However, it does result in a much higher patient dose. The comparison of the radiation dose for the same patient during the breast monitoring showed that it is absolutely necessary to acquire the breast images in the manual mode according to the settings recommended by the manufacturer if the Eklund maneuver cannot be performed. In this case, the radiation dose was lower, and the quality of the images was acceptable to the radiologist. Note that when the Eklund maneuver was possible, the radiation dose was the lowest because the AEC sensor was positioned on the breast tissue out of the BI. This low dose of radiation resulted in a higher noise level and therefore a lower SNR_BT_ than when using the recommended acquisition parameters. This acquisition condition reproduced the standard conditions of mammography and was therefore used as a reference for the phantom study. 

In the clinical example, a silicone-gel-filled breast implant was used. Silicone gel is one of the most radiopaque materials used in breast implants. A comparison of the radiopacity of this type of implant with the copper plate included in the phantom showed that silicone gel and copper materials attenuate many X-rays. However, in order to determine a radiopacity comparable to that of the copper plate included in the phantom, further research would be needed to define the exact physical properties required for silicone gel. And beyond the scope of this study with its focus on silicone implants, it would be necessary to physically characterize the level of radiopacity of different implant materials used in breast reconstruction [14,15]. Manufacturers of breast implants do not provide this information, which, depending on the type of implant, can have a very detrimental effect on diagnostic mammography. 

Two image acquisition situations were tested on the phantom. The first was the simulation of the Eklund maneuver as for the LCC projection. The phantom study simulated this case by positioning the AEC sensor placed outside the Cu plate. The image metrics and dose data obtained represent the reference values to be achieved, similarly to the clinical case when the AEC sensor was out of the BI. The reference AGD values between the clinical case and the phantom were close but were still −14% lower for the phantom. The second image acquisition situation was the simulation of a partial overlap of the AEC sensor with the BI by placing the AEC sensor below the Cu plate. The results showed that the images obtained with the AEC sensor overlapping with the Cu plate gave higher values in terms of *SDNR* and detectability, but at the cost of a much higher dose. The study highlighted the need to position the AEC cell outside the Cu plate for dose reduction. Similar trends had been observed clinically for the SNR_BT_ and AGD values as a function of the position of the AEC sensor relative to the BI. When the BI covers the whole breast, the manufacturer recommends to switch to manual mode. By using the recommended settings, similar AGD values were achieved between the clinical case and the phantom, i.e., approximately 1.5 mGy in both cases. Therefore, the IAEA phantom may be useful to reproduce clinical situations with the presence of BIs and to help identify wrong automatic settings. 

As the IAEA phantom could be applied to reproduce BI issues successfully, further investigations were carried out to evaluate the use of the IAEA methodology for the improvement of exposure settings. In the literature, Silva et al. [9] evaluated image scoring regarding the number of masses, microcalcifications and fibers by varying five mAs values on a phantom. Their study was performed using screen-film mammography, and only a visual approach was considered. In our study, three mAs values were compared: first, the recommended mAs; second, the reduced mAs by decreasing the recommended mAs using an option available on the acquisition console; and third, the reference mAs defined in automatic mode when the AEC sensor was outside the Cu plate. The results showed that the radiation dose was reduced by −13% when the reduced mAs values were used, compared to the recommended mAs values, but without a significant loss of image quality. All image metrics were determined automatically by the IAEA software rather than visually as in previous studies [8,9]. The comparison between the reduced and reference settings showed that all image quality metrics and radiation doses were significantly different. It would be possible to further reduce mAs, verify the image quality metrics at each step and validate these new acquisition conditions on patients. Therefore, the IAEA phantom may be useful in adjusting the manufacturer’s recommended exposure settings prior to its use on patients. 

The clinical case confirmed the importance of training technologists specifically for the imaging of patients with BIs as recommended by the Food and Drug Administration [52]. In fact, the lack of knowledge about how to acquire images in the specific case of breasts with BIs led to an over-exposure that could have been avoided. It highlighted the need for guidelines and expertise in implant imaging to ensure the safety of BI mammography. Patients with BIs should be referred to facilities with such expertise for their breast screening or follow-up after surgery, and technologists must be specifically trained for BI imaging. To the best of our knowledge, no incident related to radiation dose has been reported as an adverse event occurring during mammograms in women with BIs [53]. 

This study has some limitations. First, only one breast of a clinical case without lesions or calcifications was examined. Therefore, an evaluation of the detection performance of the phantom and its possible correlation with the clinical diagnosis was not possible. However, a close agreement between the clinical detection of simulated microcalcifications and the imaging performance of the 0.1 mm and 0.25 mm discs has already been demonstrated by Warren et al. [43]. It is important to note that the main focus of our study was on phantom experiments rather than clinical cases. We decided, however, to include this particular patient to illustrate the clinical acquisition techniques of mammograms of BIs. Second, this phantom study should be the subject of further trials with variable PMMA thicknesses and breast densities. In fact, only one standard breast thickness and density have been simulated. In the literature, Daskalaki et al. [8] compared several breast thicknesses in the presence of silicone breast implants based on a Monte Carlo simulation with homogeneous breast tissue. Indeed, the Monte Carlo simulation approach is very useful for quickly testing different breast implant compositions, breast thicknesses and breast tissue heterogeneities. However, as the authors pointed out, experimental work is still needed to verify and validate the results of these simulations under clinical conditions on physical phantoms as a replica of the software models of the breasts. Third, our study was limited by having only one unit involved. Nevertheless, the correlation between physical image quality parameters between system brands and/or between individual systems of the same brand and type may be scattered due to heel and geometric effects. Indeed, the largest effects may be observed in the part of the image where the Al square used to calculate the *SDNR* and d’ is positioned. The AEC sensitivity study showed that the spatial distribution of the SNR was not uniform, which meant that the physical blurring process was not constant across the detector. This change in SNR as a function of ROI position implied that the stationarity of the noise statistics within the image was limited. Careful attention should be given to objective measurements, and there are limitations on our ability to compare mammography units. An alternative would be to place the Al square in a more anatomically relevant position close to the AEC sensors. This would better characterize the portion of the image field and the receptor where the breast tissue would be located, allowing for a comparison between mammography devices and manufacturers. Fourth, this study did not evaluate the detection of low-contrast masses in the presence of breast implants, especially when varying kVp values. Also, the X-ray field used for the estimation of the AGD does not include the variation in air kerma incident upon a standard breast due to the heel effect [54]. In addition, there is still a need for a model which can adequately estimate the radiation dose to patients with BIs. 

## 5. Conclusions

To the best of our knowledge, this is the first phantom study to compare the image quality in combination with AGD in the presence of a radiopaque material simulating an implant between different image acquisition techniques. The IAEA phantom and procedures could be used for the reproduction and further understanding of inappropriate acquisition techniques in clinical situations and for the evaluation of manufacturer recommendations in manual mode. As the frequency of mammograms in patients with BIs has increased significantly in recent years, more work is required in the near future that includes more clinical cases, breast implants with different compositions and more evaluators. We hope to raise awareness on the importance of appropriate mammography techniques for patients with breast implants by using the IAEA phantom and procedure in training sessions.

## Figures and Tables

**Figure 1 bioengineering-11-00884-f001:**
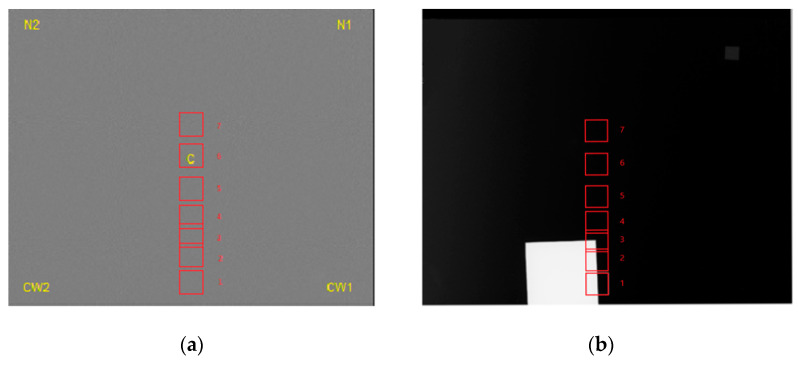
Representation of the seven AEC sensor positions labeled 1 through 7 and marked with red squares on the images of PMMA (**a**) and IAEA (**b**) phantoms. C, N1, N2, CW1 and CW2 indicate the positions of the five regions of interest used for signal-to-noise measurements.

**Figure 2 bioengineering-11-00884-f002:**
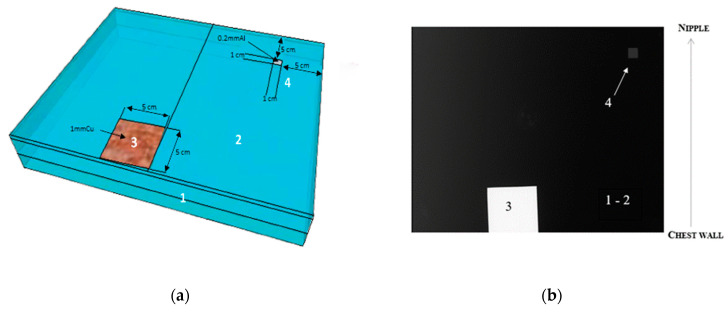
(**a**) IAEA phantom and (**b**) mammographic image, with 1–2 indicating the uniform attenuator of poly(methyl methacrylate), 3 indicating copper plate and 4 indicating aluminium foil.

**Figure 3 bioengineering-11-00884-f003:**
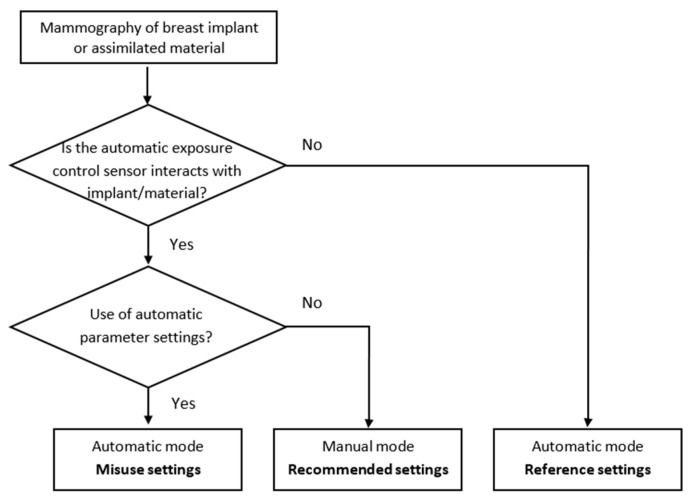
Flow chart showing the different methods of image acquisition.

**Figure 5 bioengineering-11-00884-f005:**
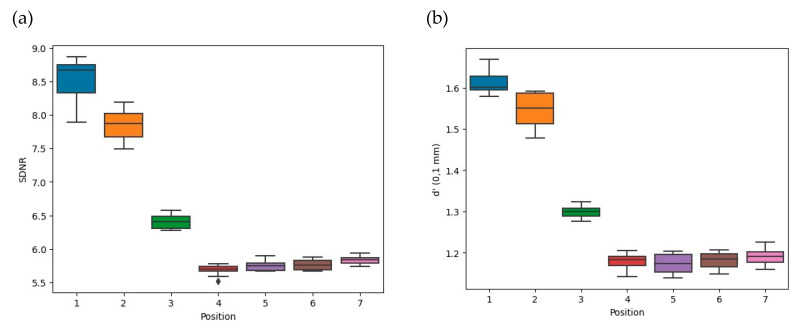
Influence on image quality, radiation dose and acquisition parameters of the seven positions of the AEC sensor labeled from 1 to 7. (**a**) Signal Difference to Noise Ratio, (**b**,**c**) Detectability index at 0.1 mm and 0.25 mm, (**d**) Tube current-time product, (**e**) Average Glandular Dose and (**f**) kilovoltage peaks.

**Figure 6 bioengineering-11-00884-f006:**
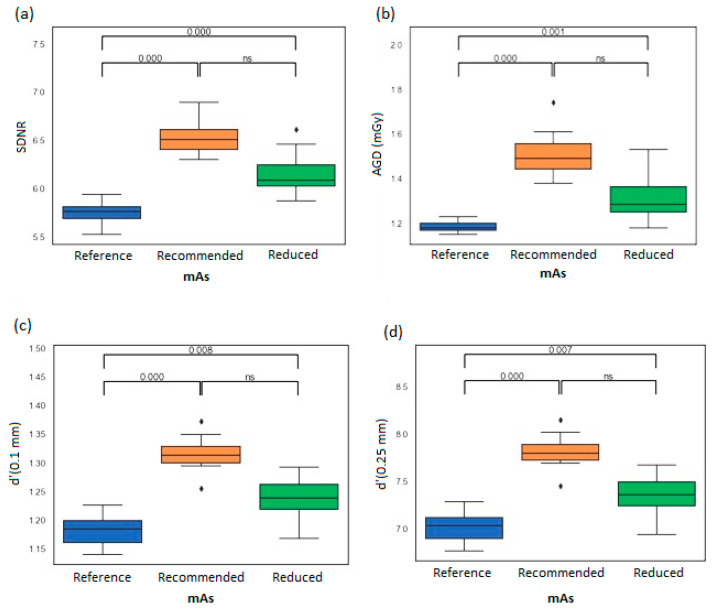
Variation in phantom image quality parameters for three different tube current-time product levels: reference, manufacturer’s recommended and reduced. (**a**) Signal- Difference-to-Noise Ratio, (**b**) Average Glandular Dose, and (**c**,**d**) values for the detectability at 0.1 mm and 0.25 mm respectively.

**Table 1 bioengineering-11-00884-t001:** Acquisition parameters, average glandular doses (AGDs) and signal-to-noise ratio of breast tissue (SNR_BT_) in manual and automatic modes. The cranio caudal (CC) and medio-lateral oblique (MLO) projections for the left (L) breast are included.

	LCC Projection with Eklund Maneuver		LMLO Projection	
	Manual-Mode Recommended Settings	Automatic-Mode Reference Settings	Manual-Mode Recommended Settings	Automatic-Mode Misused Settings
kVp	28	29	28	32
mAs	120	108	120	159
Breast thickness (mm)	46	50	51	66
Compression Force (N)	20.1	63.6	38.4	54.7
Anode/Filter	W/Rh	W/Rh	W/Rh	W/Rh
SNR_BT_	12.1 ± 6.2	4.6 ± 1.7	8.7 ± 6.4	17.2 ± 10.2
AGD (mGy)	1.7	1.4	1.4	2.5

**Table 2 bioengineering-11-00884-t002:** Comparison of criteria scoring between manual and fully AEC modes for cranio caudal (CC) and medio-lateral oblique (MLO) projections for the clinical case.

Criteria	LCC Projectionwith Eklund Maneuver	LMLO Projection
	Manual-ModeRecommended Settings	Automatic-Mode Reference Settings	Manual-ModeRecommended Settings	Automatic-Mode Misused Settings
Positioning
Breast centrally placed				
Visualization of retroglandular adipose tissue			NA	NA
Inframammary angle clearly demonstrated	NA	NA		
Full visualization of inferior breast tissue	NA	NA		
Pectoral muscle visualized	NA	NA	NA	NA
Medial border of the breast included on the image			NA	NA
Nipple in the midline (+/−10°)			NA	NA
Nipple in profile or transected by skin	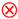			
Visibility of implant edge in the image	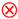			
Maximum “retropulsion” of the implant			NA	NA
Artefacts
No skin folds			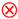	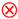
No artefacts				
Skin edges visualized			NA	NA
Sharpness
Spread of breast tissue to differentiate adipose from fibroglandular tissue				
Sharpness of glandular tissue				
Appropriate contrast				


: Correct; 
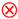
: Incorrect; NA: not applicable.

**Table 3 bioengineering-11-00884-t003:** Variation in acquisition parameters, signal-to-noise ratio (SNR) and dose values with AEC sensor positions on homogeneous PMMA.

	Position of the AEC Sensor							All AEC Positions	COV
	1	2	3	4	5	6	7		
kVp	28 ± 0	28 ± 0	28 ± 0	28 ± 0	28 ± 0	28 ± 0	28 ± 0	28 ± 0	0.00
mAs	94.6 ± 0.8	94.2 ± 0.4	93.8 ± 0.4	93.8 ± 0.4	94.0 ± 0.0	93.8 ± 0.4	94.4 ± 0.5	94.1 ± 0.2	0.01
Compressed thickness (mm)	42.2 ± 0.4	42.2 ± 0.4	42.2 ± 0.4	42.0 ± 0.0	42.0 ± 0.0	42.0 ± 0.0	42.0 ± 0.0	42.1 ± 0.1	0.01
AGD (mGy)	1.3 ± 0.0	1.3 ± 0.0	1.2 ± 0.0	1.3 ± 0.0	1.3 ± 0.0	1.2 ± 0.0	1.3 ± 0.0	1.3 ± 0.0	0.01
SNR_C_	55.4 ± 0.7	54.8 ± 0.6	54.6 ± 0.2	54.7 ± 0.5	54.7 ± 0.4	54.8 ± 0.3	54.7 ± 0.1	54.8 ± 0.2	0.01
SNR_CW1_	57.8 ± 1.0	57.4 (0.8)	56.9 ± 0.2	56.9 ± 0.1	57.0 ± 0.4	56.8 ± 0.2	56.7 ± 0.2	57.1 ± 0.2	0.01
SNR_N1_	39.8 ± 1.0	39.5 (0.3)	39.2 ± 0.4	39.3 ± 0.2	39.3 ± 0.3	39.2 ± 0.3	38.7 ± 0.2	39.3 ± 0.2	0.01
SNR_N2_	41.7 ± 1.2	41.0 (0.5)	40.7 ± 0.3	40.9 ± 0.1	40.7 ± 0.2	40.2 ± 0.5	39.9 ± 0.4	40.7 ± 0.3	0.02
SNR_CW2_	60.4 ± 1.0	58.7 (0.9)	58.2 ± 0.2	58.2 ± 0.4	58.4 ± 0.4	58.2 ± 0.2	58.2 ± 0.2	58.5 ± 0.2	0.01

Abbreviations: C: Center; CW: Chest wall; N: Nipple; COV: Coefficient of variation.

**Table 4 bioengineering-11-00884-t004:** Variation in image quality parameters and radiation dose values with the position of the AEC sensor relative to the copper plate of the phantom in automatic mode.

Automatic Mode	*SDNR*	d’ (0.1 mm)	d’ (0.25 mm)	AGD (mGy)
Misused settings	7.9 (1.8)	1.6 (0.3)	9.2 (1.7)	2.2 (0.8)
Reference settings	5.8 (0.1)	1.2 (0.0)	5.8 (0.1)	1.2 (0.0)
*p* value	<0.03 ^1^	<0.03 ^1^	<0.03 ^1^	<0.03 ^1^

^1^ Significant *p*-value of difference between misused and reference settings. Data are presented as median (interquartile range).

## Data Availability

Data are available from the authors upon request.

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
