# Peer review of "Two-Dimensional Mammography Imaging Techniques for Screening Women with Silicone Breast Implants: A Pilot Phantom Study"

_bioengineering, 2024, doi:10.3390/bioengineering11090884_

Round 1

Reviewer 1 Report

Comments and Suggestions for Authors

See attached word file.

Comments on the Quality of English Language

See attached word file.

Author Response

Reviewer 1:

General comment:

“Undoubtedly, this endeavor contributes to communities worldwide. Herein lie a few proposals and suggestions that, if included, could elevate its caliber.”

Response:

We thank the reviewer for his positive feedback of our study and we thank him very much for taking the time to review our manuscript. We thank him for the comments and suggestions to improve the manuscript. We have made significant changes to the structure of the manuscript.

Comments 1:

“Shorten title of the study. “

Response 1:

We have followed the advice of the reviewer and propose this new shortened title:

“2D mammography imaging techniques for screening women with silicone breast implants: a pilot phantom study”

Comments 2:

“Remove the words ‘Background’, ‘Methods’, ‘Results’, and ‘Conclusions’ along with full colon “:” from abstract.”

Response 2:

Thank you for pointing this out. The words ‘Background’, ‘Methods’, ‘Results’, and ‘Conclusions’ have been removed from the abstract

Comments 3:

“Abstract has not been written professionally. Authors are requested to rewrite it in a more compact way, without any loss of generality. While rewriting the abstract, clearly highlight current issues as to why this study is necessary, briefly describe your strategy as to how did you resolve the issue, and finally highlight significance and usefulness of your scheme.”

Response 3:

We thank the reviewer for pointing out the content of the abstract. We have completely rewritten it and hope that it now reflects our study.

Comments 4:

“Break long paragraphs into smaller ones, see first paragraphs of Section1 and Section 2.3. “

Response 4:

We thank the reviewer for this comment. We have restructured the Materials and Methods section into sub-paragraphs to clarify understanding of the manuscript. We have also added a flowchart in Figure 3 to facilitate reading.

Comments 5:

“Replace star-sign in 24*29 by × using MathType or Equation editor. “

Response 5:

We thank the reviewer and have replaced the star * by ×.

Comments 6:

“Break long sentences into smaller ones, with clarity. For example, “The 2D mammograms of a 47-year-old woman with a history of right triple negative grade 3 breast cancer, hormone receptors and HER2 negative treated by total mastectomy, radiation therapy and hormone therapy and bilateral breast reconstruction with anatomical BI were retrospectively studied.””

Response 6:

We thank the reviewer for this remark. Indeed, some sentences were too long. We have reviewed the entire publication to try and shorten certain sentences and split into small paragraphs. For example we replaced the cited sentence by :

« Two 2D mammograms of a 47 year-old woman were retrospectively studied. The patient underwent her annual follow-up after breast reconstruction in 2021 and 2022.  This patient has a history of grade 3 right triple negative breast cancer, hormone receptors and HER2 negative. She was treated by a total and partial mastectomy of her right and left breasts respectively, radiation therapy and hormone therapy. »

Comments 7:

“ “It was composed of a 24 × 30 × 1684.5 cm”. (It is ??3 ). Similarly, check for other typos.”

Response 7:

We thank the reviewer and have modified the text as suggested.

Comments 8:

 “There are numerous cases of connected words, e.g., “Mammographicimage ”. Please insert proper spaces to avoid confusion.”

Response 8:

We thank the reviewer for this comment and have carefully read the manuscript to add the spaces.

Comments 9:

“Enhance contrast of the image in Figure 1(a).”

Response 9:

We thank the reviewer and have modified the color of the background of the Figure 1a in order to improve the contrast.

Comments 10:

 “Shorten title of Table-1. Do the same for other tables.”

Response 10:

Thank you for pointing this out. The titles of Table 1 and Table 4 have been shortened.

Comments 11:

 “Also, use short description/title/legend for each figure and then explicitly explain images/figures in the following text.”

Response 11:

We agree with this comment. Accordingly, we have changed legends and titles to shorten them.

Comments 12:

 “Revisit Table-3, check whether the captions given in the top left corner are placed correctly or not.”

Response 12:

We thank the reviewer for his vigilance. One line in Table 3 was duplicated. We have deleted it.

Comments 13:

 “The contrast of the images given in Figure 4 & 5 can be further improved.”

Response 13:

We thank the reviewer for this comment. The contrast of these two figures has been increased.

Comments 14:

 “Finally, pay attention to grammar, language, and formatting issues”

Response 14:

We thank the reviewer for this advice. The publication was proofread by an English-speaking person.

Reviewer 2 Report

Comments and Suggestions for Authors

1. What are the main contributions of this article, how does this differ from existing approaches, and what are the advantages?

2. How is Eq. 2 solved? Can you provide detailed steps to solve it?

3. Some related work should be introduced:BDAL: Balanced Distribution Active Learning for MRI Cardiac Multistructures Segmentation, ALVLS: Adaptive local variances-Based levelset framework for medical images segmentation

4. You can provide more experimental results to verify the validity of the proposed method.

Author Response

Comments 1:

“What are the main contributions of this article, how does this differ from existing approaches, and what are the advantages?”

Response 1:

We would like to thank the reviewer for its request for details on the main contributions of the study. The structure of the manuscript has been revised with more concise paragraphs to clarify the different sections and hopefully give the reader a better understanding of the subject.

About the contributions of the study: There are currently no mammography studies using a test object including radiopaque material to assess image quality and the radiation dose delivered in presence of breast implant. Our study aimed first, to evaluate the impact on image quality and radiation dose of three 2D mammographic acquisition techniques in the presence of silicone breast implants. Therefore, we have illustrated the three data acquisition techniques using a clinical case. We then reproduced these techniques on a new test object developed by the IAEA. Image metrics were quantified to compare the impact of the different acquisition techniques on image quality and radiation dose.

The simulation of different acquisition conditions on the test object will raise awareness in the medical and paramedical community of the specific care required for patients with breast implants. The test object can easily be used in training sessions. We have also shown that the manufacturer's recommendations for acquisition parameters can be adjusted. The test object and associated image processing software can be used to automatically calculate image metrics. This makes it easy to compare image performance between different acquisition parameters.

Comments 2:

“2. How is Eq. 2 solved? Can you provide detailed steps to solve it?”

Response 2:

We thank the reviewer for his question. The main steps in calculating equation 2 are as follows:

A non-pre-whitening model observer with eye filter (NPWE) was used to calculate the detectability index (d’NPWE) (Eq. 2):

where u is the spatial frequency in a visual transfer function (VTF), C is the contrast measured using the Al square, S is the Fourier transform of a disk, MTF is the modulation transfer function of the detector before sampling, nNPS is the normalized noise power spectrum for the image of interest.

  • Normalized Noise Power Spectrum (nNPS) was estimated from a 512 × 512 region in a homogeneous area of the phantom image. Half-overlapping ROIs of 256 × 256 pixels were then extracted for the calculation of the 2D NPS. Using seven spatial frequency bins on either side of the axes of the axial spectra, the axial and radial NPS curves were sectioned from the 2D spectra. The nNPS was calculated using a standard formula.
  • The presampled Modulation Transfer Function (MTF) was measured from the edges of the Cu plate in the phantom. The directional MTF was obtained at highly supersampled pseudo-frequencies. These were produced by the sloping horizontal and vertical edges. The two orthogonal MTF curves were then averaged and evaluated at the same frequencies as the nNPS.
  • Two task functions assumed to represent circular signals of 0.25 mm and 0.1 mm diameters were simulated, as a reasonable approximation of the smallest microcalcifications. This is a typical and important task for mammography imaging systems. The assumption was that the circular object would have the same contrast as the Al-square. The suitability of aluminum as a material to represent microcalcifications has been demonstrated previously.
  • The Fourier transform of this disc-shaped object with a radius of R was a Bessel function of the first order. Using a nominal image magnification of 1.5 and a viewing distance of 400 mm, the eye filter was modelled according to the VTF.

Comments 3:

“Some related work should be introduced: BDAL: Balanced Distribution Active Learning for MRI Cardiac Multistructures Segmentation, ALVLS: Adaptive local variances-Based levelset framework for medical images segmentation”

Response 3:

We would like to thank the reviewer for his suggestions for segmenting methods. These methods are interesting, but our study does not focus on the segmentation of mammography images. The automatic analysis of the images of the IAEA test object is carried out using regions of interest positioned on the aluminium, copper and background.

Comments 4:

“You can provide more experimental results to verify the validity of the proposed method..”

Response 4:

We thank the reviewer for this comment. Indeed, his request constitutes the main limitation of our study. The aim of our study was to investigate the feasibility of using the IAEA test object to simulate different mammography acquisition conditions in the presence of breast implants. It was a proof of concept with a link to a clinical case analysis. We had a PMMA thickness constraint for the IAEA test object that we didn't want to change for this study. The difficulty was therefore to identify images of a clinical case of approximately equivalent compressed breast thickness and with different image acquisition conditions. This study is intended to be the starting point for further investigations. We have also begun to evaluate different types of breast implants in terms of dosimetry.

Reviewer 3 Report

Comments and Suggestions for Authors

I am grateful for the opportunity to review the study ‘Pilot study to evaluate the IAEA procedure and phantom to 2 guide image acquisition technique in 2D mammography 3 screening of women with silicone breast implants’. This refers to the need for knowledge of specific techniques for the analysis of mammograms in patients with breast implants (BI). They used a retrospective analysis of a clinical case with silicone BI to assess the feasibility of using a new International Energy Agency (IAEA) test object containing a radiopaque copper plate to guide the acquisition technique and determine optimal settings. Methods: Image quality (IQ) and average glandular dose (AGD) were determined on a single Hologic Selenia Dimensions® unit. IQ and AGD of the left breast were determined for each 2D projection. The IAEA phantom was used to reproduce the acquisitions observed in the clinic. In fully automatic mode, the IQ and AGD of the first three positions of the automatic exposure control (AEC) sensor were compared with those of the other four positions, defined as reference values outside the copper plate. The findings indicate that the IAEA test object could be used for the reproduction of acquisition techniques in clinical situations and for the evaluation of the manufacturer's recommendations.

The study is innovative and well thought out, however, I will make some suggestions in order to make the study comprehensible to different types of readers who might be interested in the subject matter.

1. At the end of the introductory section, the authors state that the study has two objectives:

"First, to analyse the radiation exposure and image quality of a clinical case in manual and automatic settings in the presence of silicone BI.

Second, to explore whether and how IAEA tools and procedures could assist in the selection of the most appropriate X-ray beam settings in the presence of BI."

Therefore, in the results section, they should be presented according to the resolution of the stated objectives. If they need to further specify each of them, they should do so in the introduction in the form of sub-objectives or research questions.

2. For a better understanding, the Materials and Methods section can be subdivided into the following sub-sections:

1. Participants (even if it is a case design it should reflect the characteristics of the patient). It is not clear to me how many case histories were reviewed and how many patients' mammograms were used. It is also not clear to me the characteristics of the patients, even if they had a breast implant.

2. Instruments, this section should include all the instruments used.

3. Procedure, the process should be reflected and it is recommended to provide a graphic in which traceability can be followed.

4. Data analysis, this section is listed as statistical analysis, however, this should include not only the statistical tests used but also the purpose for which they have been used. In other words, they are used to test hypotheses, which is why it is recommended to include them at the end of the introduction and in this section to include the tests that are going to be used to test each one of them.

5. The results section should present the results following an order according to the hypotheses or research questions raised.

6. The discussion section should include the relationship between what was proposed in the introduction in the state of the art and what was found in the study. No citations from those included in the introduction have been included.

7. The conclusions section should be more extensive and should include data on the limitations of the study and on future lines of research.

8. One relevant aspect is that the authors do not provide data on the ‘Institutional Review Board Statement’, stating that it is not applicable, and it is applicable because they work with data on highly sensitive individuals. They should provide the authorisation report to carry out the study by the institution that supports this research.

9. In the ‘Informed Consent Statement’ section, they must include that the consent of the participants was obtained in writing.

10. As this is such a current topic, the references presented by the authors in the last five years are scarce (23.40%), with none from the year 2024. Therefore, the number of citations should be increased, especially for the years 2023 and 2024.

11. Check that the way references are cited follows the journal's standards.

Author Response

General comment:

““I am grateful for the opportunity to review the study ‘Pilot study to evaluate the IAEA procedure and phantom to 2 guide image acquisition technique in 2D mammography 3 screening of women with silicone breast implants’. This refers to the need for knowledge of specific techniques for the analysis of mammograms in patients with breast implants (BI). They used a retrospective analysis of a clinical case with silicone BI to assess the feasibility of using a new International Energy Agency (IAEA) test object containing a radiopaque copper plate to guide the acquisition technique and determine optimal settings. Methods: Image quality (IQ) and average glandular dose (AGD) were determined on a single Hologic Selenia Dimensions® unit. IQ and AGD of the left breast were determined for each 2D projection. The IAEA phantom was used to reproduce the acquisitions observed in the clinic. In fully automatic mode, the IQ and AGD of the first three positions of the automatic exposure control (AEC) sensor were compared with those of the other four positions, defined as reference values outside the copper plate. The findings indicate that the IAEA test object could be used for the reproduction of acquisition techniques in clinical situations and for the evaluation of the manufacturer's recommendations.

The study is innovative and well thought out, however, I will make some suggestions in order to make the study comprehensible to different types of readers who might be interested in the subject matter.

Response:

We thank the reviewer for his positive feedback of our study and we thank him very much for taking the time to review our manuscript. We thank him for the comments and suggestions to improve the manuscript. We hope that the major structural changes to the manuscript will meet his expectations.

Comments 1:

“At the end of the introductory section, the authors state that the study has two objectives:

"First, to analyse the radiation exposure and image quality of a clinical case in manual and automatic settings in the presence of silicone BI.

Second, to explore whether and how IAEA tools and procedures could assist in the selection of the most appropriate X-ray beam settings in the presence of BI."

Therefore, in the results section, they should be presented according to the resolution of the stated objectives. If they need to further specify each of them, they should do so in the introduction in the form of sub-objectives or research questions.”

Response 1:

We'd like to thank the reviewer for this comment, as the objectives were not precise enough when compared with the overall results presented. We have therefore completed the objectives at the end of the introduction section:

“First, to illustrate the different techniques of 2D mammography image acquisition in presence of BI on a clinical case. Second, to evaluate how silicone BI radiopacity affects image quality and radiation dose depending on technique. Third, to explore whether and how IAEA tools and procedures could assist in the selection of the most appropriate X-ray beam settings, by simulating and comparing the different acquisition techniques.”

Comments 2:

“For a better understanding, the Materials and Methods section can be subdivided into the following sub-sections:”

Response 2:

We thank the reviewer for this request to restructure the various paragraphs in order to improve the clarity of the manuscript. We have restructured the Materials and methods section into smaller paragraphs, taking into account the order of topics proposed by the reviewer. All modifications on pages 3 to 7 appear underlined in the manuscript.

Comments 3:

“Participants (even if it is a case design it should reflect the characteristics of the patient). It is not clear to me how many case histories were reviewed and how many patients' mammograms were used. It is also not clear to me the characteristics of the patients, even if they had a breast implant.”

Response 3:

We agree with the reviewer and, as part of the restructuring of the Materials and Methods section, we have clarified the information on the clinical case, including the number of mammograms analyzed retrospectively.

Comments 4:

“Instruments, this section should include all the instruments used.”

Response 4:

We have followed the reviewer's recommendations and described in the following order the mammography Unit, AEC sensor, and phantoms.

Comments 5:

“Procedure, the process should be reflected and it is recommended to provide a graphic in which traceability can be followed.”

Response 5:

We agree that understanding the different acquisition methods could be made easier with the help of a graph. We therefore added a flowchart to the manuscript named Figure 3.

Comments 6:

“Data analysis, this section is listed as statistical analysis, however, this should include not only the statistical tests used but also the purpose for which they have been used. In other words, they are used to test hypotheses, which is why it is recommended to include them at the end of the introduction and in this section to include the tests that are going to be used to test each one of them.”

Response 6:

We thank the reviewer for pointing out this lack of information in the statistical study. We have supplemented the paragraph with the following information: “A statistical approach was used for the interpretation of the image metrics obtained from the IAEA phantom with the different acquisition methods. First, to compare the impact of AEC sensor position according to reference and misuse settings on image quality. Second, to compare the image quality performance achieved using the recommended and optimized mAs manual settings, with that of the reference automatic mAs setting.”

Comments 7:

“The results section should present the results following an order according to the hypotheses or research questions raised.”

Response 7:

We would like to thank the reviewer and have revised the structure with paragraphs and headings; we hope they are better suited to the objectives of the manuscript.

Comments 8:

“The discussion section should include the relationship between what was proposed in the introduction in the state of the art and what was found in the study. No citations from those included in the introduction have been included.”

We thank the reviewer for this comment. We revised this point.

Comments 9:

“The conclusions section should be more extensive and should include data on the limitations of the study and on future lines of research.”

We would like to thank the reviewer and we have completed the conclusion on future research.

Comments 10:

“One relevant aspect is that the authors do not provide data on the ‘Institutional Review Board Statement’, stating that it is not applicable, and it is applicable because they work with data on highly sensitive individuals. They should provide the authorisation report to carry out the study by the institution that supports this research.”

We thank the reviewer for this request for clarification. This clinical case study is a non-interventional study because it was conducted retrospectively. We used the CARE checklist of information to include when writing a case report. In this specific case of a case report, the statement of the institutional review board is not requested.

Comments 11:

“In the ‘Informed Consent Statement’ section, they must include that the consent of the participants was obtained in writing.”

The radiologist in charge of this patient's medical follow-up in our department had her spoken and written consent to use images of her left breast for this research. I have enclosed on the JCM website the information letter given to the patient, specifying her right to withdraw if she did not wish to take part in the study.

Comments 12:

“As this is such a current topic, the references presented by the authors in the last five years are scarce (23.40%), with none from the year 2024. Therefore, the number of citations should be increased, especially for the years 2023 and 2024.”

We have followed the reviewer's recommendations and updated our list of references for the last two years. Recently published studies focus primarily on screening and breast reconstruction techniques.

Comments 13:

“Check that the way references are cited follows the journal's standards.”

We thank the reviewer for this comment. To compile our bibliography, we use zotero software and have set it up to comply with the citation style of the "Multidisciplinary Digital Publishing Institute".

Round 2

Reviewer 2 Report

Comments and Suggestions for Authors

1. The figures lacks clarity and needs improvement.

2. Performance comparisons with the latest methods need to be added to verify the effectiveness of the proposed method.

3. The manuscript format is too confusing. Careful revision is recommended.

Author Response

Comments 1:

“The figures lacks clarity and needs improvement.”

Response 1:

We thank the reviewer for this comment. We have modified the color of the background of the Figure 1a in order to improve the contrast. The contrast of the images in Figures 5 and 6 has been increased and the text on the abscissa and ordinate has been made larger.

Comments 2:

“Performance comparisons with the latest methods need to be added to verify the effectiveness of the proposed method”

Response 2:

We thank the reviewer for his suggestion. We added these sentences in the “Discussion” section:

L504-506: “All processing of the phantom images were done automatically by the software developed by the IAEA. This means better reproducibility of results compared to visual analysis [9], which may be subject to inter- and intraobserver variabilities [8].

.”

L550-553: “In the literature, Silva et al. [9] evaluated image scoring regarding number of masses, microcalcifications and masses by varying five mAs values on phantom. This study was performed in screen-film mammography. Only a visual approach was considered.”

L561-562: “All image metrics were determined automatically by the IAEA software rather than visually as in previous studies [8,9].”

L586-592: “In the literature, Daskalaki et al. [8] compared several breast thicknesses in the presence of silicone breast implant based on a Monte Carlo simulation with homogeneous breast tissue. Indeed, the Monte Carlo simulation approach is very useful for fast testing of different breast implant compositions, breast thicknesses and breast tissue heterogeneities. However, as the authors pointed out, experimental work is still needed to verify and validate the results of these simulations under clinical conditions on physical phantoms as a replica of the software models of the breasts.”

Comments 3:

“The manuscript format is too confusing. Careful revision is recommended.”

Response 3:

We would like to thank the reviewer for this remark. This lack of clarity in the structure was also pointed out by one of the reviewers. In fact, the paragraphs were too long and the order of the different sections could make the manuscript difficult to read. The structure of the manuscript has been totally revised with more concise paragraphs to clarify the different sections and hopefully give the reader a better understanding of the subject.

For a better understanding, the “Materials and Methods” section was subdivided into the following sub-sections:

  1. Clinical case illustrating the different acquisition techniques.
  2. Digital mammography system
  3. Characterization of the automatic exposure control
  4. Description of the IAEA phantom for mammography
  5. Description of 2D mammographic imaging techniques for breast implants and assimilates
  6. Image quality evaluation of breast implants and assimilates
  7. Data analysis

The structure of the “Results” section was also modified according to the objectives of the manuscript described at the end of the introduction:

“First, to illustrate the different techniques of 2D mammography image acquisition in presence of BI on a clinical case. Second, to evaluate how silicone BI radiopacity may affect image quality and radiation dose depending on technique. Third, to explore whether and how IAEA tools and procedures could assist in the selection of the most appropriate X-ray beam settings by simulating and comparing the different acquisition techniques.”

Therefore the structure of the “Results” section was modified as follows:

  1. Clinical case illustrating the different acquisition techniques
    1. Impact of breast implant radiopacity according to acquisition techniques based on the recommended settings compared to the reference settings
    2. Impact of breast implant radiopacity according to acquisition techniques based on the recommended settings compared to the misused settings
  2. Digital mammography system: Tests of automatic exposure control sensitivity
  3. Replication of clinical acquisition techniques with the IAEA phantom
    1. Comparison of acquisition techniques in automatic mode: reference versus misused settings with a varying position of the AEC sensor
    2. Comparison of several settings in manual mode

Round 3

Reviewer 2 Report

Comments and Suggestions for Authors

none.